# Diversity and Communities of Fungal Endophytes from Four *Pinus* Species in Korea

Soon Ok Rim [1], Mehwish Roy [1], Junhyun Jeon [1], Jake Adolf V. Montecillo [1], Soo-Chul Park [2] and Hanhong Bae [1,*]

[1] Department of Biotechnology, Yeungnam University, Gyeongsan 38541, Gyeongbuk, Korea; sorim@ynu.ac.kr (S.O.R.); mehwishroy@gmail.com (M.R.); jjeon@yu.ac.kr (J.J.); adolfjake@ynu.ac.kr (J.A.V.M.)

[2] Crop Biotechnology Institute, Green Bio Science & Technology, Seoul National University, Pyeongchang 25354, Kangwon, Korea; scpark1@snu.ac.kr

* Correspondence: hanhongbae@ynu.ac.kr; Tel.: +82-53-810-3031

**Abstract:** Fungal endophytes are ubiquitous in nature. They are known as potential sources of natural products, and possible agents for biocontrol attributing to their ability to produce a repertoire of bioactive compounds. In this study, we isolated fungal endophytes from three different tissues (needle, stem and root) of four *Pinus* species (*Pinus densiflora*, *Pinus koraiensis*, *Pnus rigida*, and *Pinus thunbergii*) across 18 sampling sites in Korea. A total number of 5872 culturable fungal endophytes were isolated using standard culturing techniques. Molecular identification based on the sequence analyses of the internal transcribed spacer (ITS) or 28S ribosomal DNA revealed a total of 234 different fungal species. The isolated fungal endophytes belonged to Ascomycota (91.06%), Basidiomycota (5.95%) and Mucoromycota (2.97%), with 144 operational taxonomic units (OTUs) and 88 different genera. In all sampling sites, the highest species richness (S) was observed in site 1T (51 OTUs) while the lowest was observed in site 4T (27 OTUs). In terms of diversity, as measured by Shannon diversity index (H′), the sampling site 2D (H′ = 3.216) showed the highest while the lowest H′ was observed in site 2K (H′ = 2.232). Species richness (S) in three different tissues revealed that root and needle tissues are highly colonized with fungal endophytes compared to stem tissue. No significant difference was observed in the diversity of endophytes in three different tissues. Among the four *Pinus* species, *P. thunbergii* exhibited the highest species richness and diversity of fungal endophytes. Our findings also revealed that the environmental factors have no significant impact in shaping the composition of the fungal endophytes. Furthermore, FUNGuild analysis revealed three major classifications of fungal endophytes based on trophic modes namely saprotrophs, symbiotrophs, and pathotrophs in four *Pinus* species, with high proportions of saprotrophs and pathothrops.

**Keywords:** host specificity; fungal endophyte; fungal diversity; pine trees

## 1. Introduction

Microbial endophytes (fungi and bacteria) are microbes that live in plant tissues without eliciting any disease symptoms and thus represent symbiotic interactions between microbes and host plants [1–3]. Endophytes perform various beneficial functions in plants; they can promote plant growth, suppress pathogens and can plant tolerance to abiotic stresses [4,5]. Fungal endophytes have been known to produce numerous bioactive compounds that can be exploited in the fields of agriculture and pharmaceutical industries, as insecticidal, antimicrobial, and anticancer agents [6,7]. In addition, fungal endophytes can influence plant ecology, the evolution of plant community structure, and the diversity of interacting organisms including nematodes and insects [8–10]. For these reasons, many researchers have studied the importance of fungal endophytes' geographical distributions, as well as their ecological associations in the plant populations and community biodiversity [11–13].

*Pinus* species belong to the conifer family *Pinaceae*, the largest conifer family with approximately 120 species worldwide. Pine trees are widely distributed in the world, and they grow well on acid and drainage soils [14]. They are also able to tolerate poorly drained wet soils. According to the number of vascular bundles in the needle, the genus can be classified into two subgenera: *Pinus* subgenus *Pinus*, and *Pinus* subgenus *Strobus*. *Pinus* subgenus *Pinus* has hard wood and has two or three needles per fascicle, while *Pinus* subgenus *Strobus* has soft wood and has five needles per fascicle [15]. Pine trees comprise over 35% of the total tree species in Korea [16]. These trees are considered to have ecological and cultural importance [17], and further observed as the most important afforestation tree species in Korea [18]. However, the coniferous forests in Korea are considered to be in a state of decline due to several factors, such as the destruction of conifer forests by humans, air pollution, acid rain, and the effects of climate change [13,19].

Previous studies have reported the isolation of fungal endophytes from many different tree species including pine trees [13,20–36]. However, most studies conducted in pine trees focused on the distribution and composition of fungal endophytes from needles [23,26,33] and found the phylum Ascomycota as the most abundant fungal endophytes from needle tissue [35]. However, fungal endophytes from pine tree in other tissues like roots and stems are still poorly explored and characterized. Thus, the distribution and composition of the fungal endophyte community associated with pine tree species in different parts (needle, stem and root) from different regions are extremely important to reveal the identity of unexplored endophytes. Several studies have demonstrated the effect of environmental and host factors in the composition of fungal endophytes inhabiting *Pinus* species. For instance, the study of Wang et al. (2007) [25] revealed the presence of *Phomopsis archeri*, *Alternaria alternate*, and *Pestalotiopsis besseyi* in stem, and *Leptostroma* sp. and *P. lingam* in needle tissues as the frequently isolated fungal endophytes of *P. tabulaeformis*, a major tree in northern Chinese forests. Moreover, it has been shown also that varying seasons and types of tissues have a significant impact on the number of isolated fungal endophytes [25]. In Spain, various pathogenic fungal endophytes such as *Davidiella tassiana*, *Ulocladium* sp., *Epicoccum nigrum*, *Phoma herbarum*, *Pleospora herbarum*, *Aureobasidium pullulans,* and *Phaeomoniella* sp. were isolated in one of the most abundant conifers, *P. halepensis* [35]. The data on the isolated fungal endophytes have been correlated to the decline of *P. halepensis* and further provided an understanding on endophyte–host interactions. On the other hand, regarding diverse salt-tolerant fungal endophytes, many of them were *Penicillium* sp., were isolated from the black pine, *P. thunbergii* in Korea [30]. To date, the identification of fungal endophytes is usually conducted using the sequences generated from the nuclear ribosomal internal transcribed spacer region (rDNA ITS) and/or from the nuclear large subunit ribosomal DNA (LSU rDNA) [37,38].

In this study, we aimed to provide an overview of the diversity and distribution of fungal endophytes from different parts (needle, stem and root) of four *Pinus* species collected from different geographical areas in Korea. We also sought to determine the specific host endophytes interactions as well as the impact of the various geographical conditions on the fungal endophyte communities associated with the four *Pinus* species. The data presented here will not just add information on the existing literature of fungal endophytes isolated from pine trees, but will serve also as basis for further understanding of endophyte–host and environment interactions.

## 2. Materials and Methods

### 2.1. Collection of Pine Tree Samples

During the summer (June–August) of 2016, pine tree samples (needle, stem and root) were collected from four *Pinus* species (*P. densiflora*, *P. koraiensis*, *P. rigida*, and *P. thunbergii*) in Korea (Supplementary Table S1). *P. densiflora* (1D–5D) and *P. thunbergii* (1T–5T) were collected from five sampling sites, while *P. koraiensis* (1K–4K) and *P. rigida* (1R–4R) were collected from four sampling sites (Figure 1). From each sampling site, six trees (six replications of 40–50 year old trees) were randomly collected.

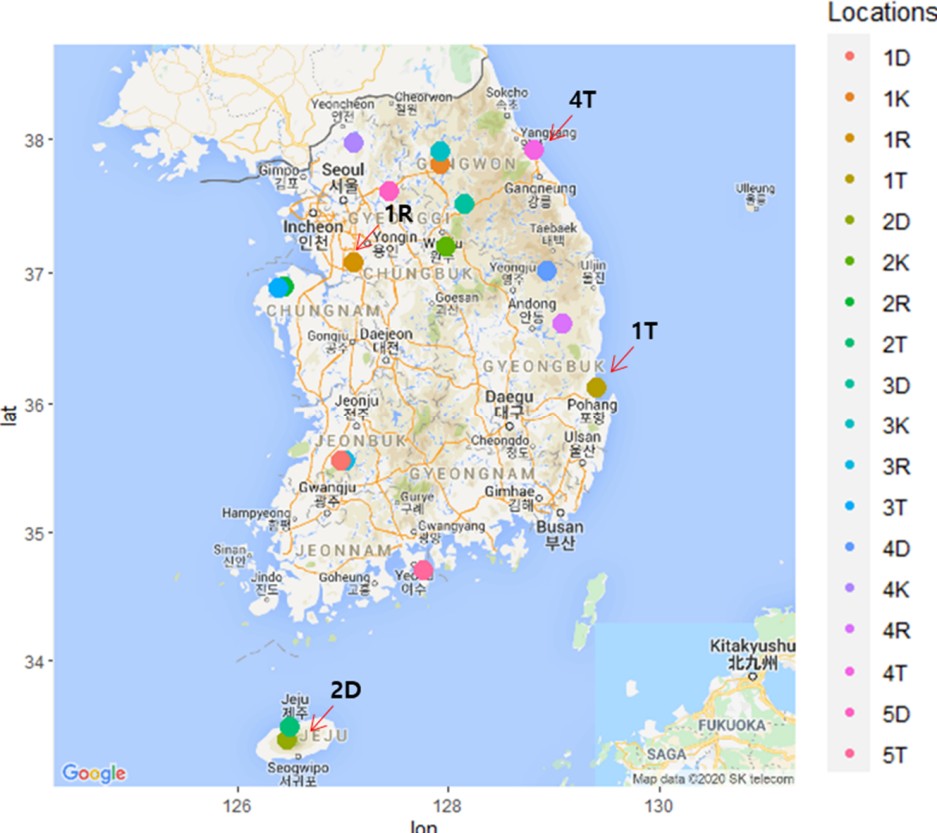

**Figure 1.** A map showing locations of 18 sampling sites in Korea. Each color circle represents sampling site labeled as 1D to 5T. Google map was modified to indicate sampling locations using ggmap package available in R programming (http://www.R-project.org, v3.6) [39].

For needle sample, branches were collected from the two opposite sides of the top of the tree. From the collected branches, the needle samples (2 year old) were harvested and cut into 0.5 cm segments. Stem samples, 1 m above the ground were collected using a sterilized increment borer. The stem samples (15–20 cm) comprised the bark, phloem, sapwood, and heartwood. The root samples were collected from 10 to 25 cm below the ground. The average length of root samples was approximately 30–50 cm.

A total of 324 samples comprising the needle, stem and root were obtained from the randomly collected 108 pine trees. All samples were put into clean zip bags separately. Samples were transported to the lab in an icebox, kept at 4 °C, and processed within 48 h. Additionally, site conditions (altitude, precipitation) and soil properties (pH, organic matter, silicic acid, K, Ca and Mg ions) data were obtained from the government online data source (http://www.kma.go.kr and http://www.rda.go.kr, accessed on 15 June 2016).

### 2.2. Isolation of Fungal Endophytes

Fungal endophytes were isolated according to previously established procedures [40]. All tissue samples (needle, stem, and root) were processed within 48 h after collection. The roots samples were rinsed thoroughly with running tap water on a 0.5 mm sieve to remove the soil. The needle samples were cut into 0.5 cm segments and the stem and root samples were cut into segments of around 1 cm each using a sterilized blade. The tissues were sequentially surface-sterilized by immersing first in 70% ethanol for 30 s, followed by 2% solution of sodium hypochlorite for 10 min, then 70% ethanol for 30 s, and finally rinsed with sterile distilled water three times for 1 min. The surface-sterilized samples were allowed to dry on sterile paper towels inside a laminar air flow bench. After drying, six to seven tissue segments were placed on a petri dish containing potato dextrose agar (PDA) with 200 mg/L ampicillin and streptomycin to inhibit bacterial growth and then

incubated at 25 °C for 2–3 weeks [41]. The tissue segments were observed every day for the appearance of fungal growth. The fungal mycelia growing out of the tissue segments were collected from the culture plate and continuously maintained on fresh PDA plates. The fungal isolates were initially identified as morpho-species. For the identification of morpho-species, fungal isolates from pure cultures were characterized based on mycelium color, structure, conidia type, and conidiophore morphology (size, color, shape, ornamentation, etc.). The identified morpho-species were then subjected to molecular identification.

### 2.3. Molecular Identification

For molecular identification, the internal transcribed spacers (ITS) or large subunit of the nuclear ribosomal DNA (LSU rDNA) were amplified from all fungal endophytes using the universal primers (ITS1 5′-TCCGTAGGTGAACCTGCGG-3′; ITS4 5′-TCCTCCGCTTATTGATATGC-3′, NL1 5′-GCATATCAATAAGCGGAGGAAAAG-3′; NL4 5′-GGTCCGTGTTTCAAGACGG-3′) [37,38] according to Bionics protocol (Seoul, Korea). The amplified sequences were assembled and analyzed using Geneious Prime v10.1.3 software (Biomatters, Auckland, New Zealand) and recognized through BLAST search in NCBI database (https://blast.ncbi.nlm.nih.gov/, accessed on 20 July 2020).

### 2.4. Data Analysis

To quantitatively infer the composition and diversity of the fungal endophytes, diversity measures such as the relative abundance, species richness (S), and Shannon diversity index (H′) were calculated. The relative abundance/proportion was estimated as a percentage of the number of fungal species, operational taxonomic units (OTUs) or phylum divided by its total number. Species richness (S) among the isolates was calculated as the number of species recovered from the particular sampling site or tree tissue. Shannon index (H′) was measured using the following equation: H′ = −Σ (Pi × In Pi), where Pi is the relative proportion of species in a specific sampling site or tree tissues [42]. Graphs were made using a GraphPad Prism 7 software (San Diego, CA, USA).

To determine the effect of various conditions on the composition of fungal community, we performed a permutational multivariate analysis of variance (PERMANOVA) and principal coordinates analysis (PCoA) on the Bray-Curtis dissimilarity indices, using the data from tissues, *Pinus* species and sites, through R packages vegan and phyloseq [43]. Specific site conditions (altitude, precipitation), soil conditions (pH, organic matter, silicic acid, K, Ca and Mg ions) and tree age were also included in the analysis (Supplementary Table S1).

FUNGuild (https://github.com/UMNFuN/FUNGuild, accessed on 11 November 2020) was used to predict the functions [44] of all fungal endophytes isolated from the needle, stem and root in four *Pinus* species. The data were statistically analyzed by performing the analysis of variance and Tukey's test ($p < 0.05$) using R (version 3.6).

## 3. Results

### 3.1. Isolation and Identification of Fungal Endophytes

We isolated a total of 5872 culturable fungal endophytes (isolates) from three different tissues of four *Pinus* species across 18 different sampling sites in Korea. Root tissue presented the largest number of fungal endophytes ($n = 2528$) followed by the needle ($n = 2381$), and the stem tissues ($n = 963$). The total number of isolated culturable fungal endophytes varied depending on *Pinus* species: *P. thunbergii* ($n = 1791$), *P. densiflora* ($n = 1469$), *P. rigida* ($n = 1379$), and *P. koraiensis* ($n = 1233$). From the 5872 fungal isolates, 234 morpho-species were identified. Further identification of the morpho-species based on rDNA ITS or LSU rDNA revealed 88 genera with a total of 144 operational taxonomic units (OTUs). The sequences were deposited in GenBank (Supplementary Table S2). The fungal endophytes belonged to three known phyla, namely Ascomycota, Basidiomycota, Mucoromycota, representing 91.06%, 5.95% and 2.97%, respectively (Figure 2A). Most of the identified species belonged to *Penicillum* (14.89%) of Ascomycota, and *Umbelopsis* (2.12%) of

Mucoromycota (Figure 2B). The most frequently occurring phylum Ascomycota, included 127 OTUs. Basidiomycota group was represented by 13 different OTUs. Mucoromycota was represented by four OTUs.

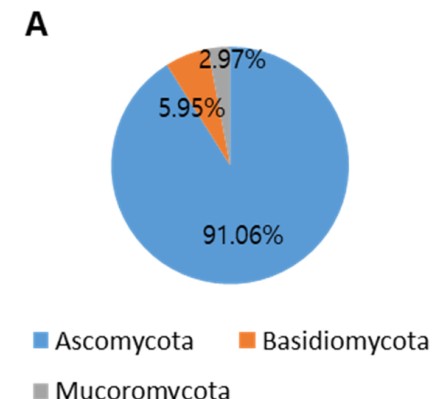

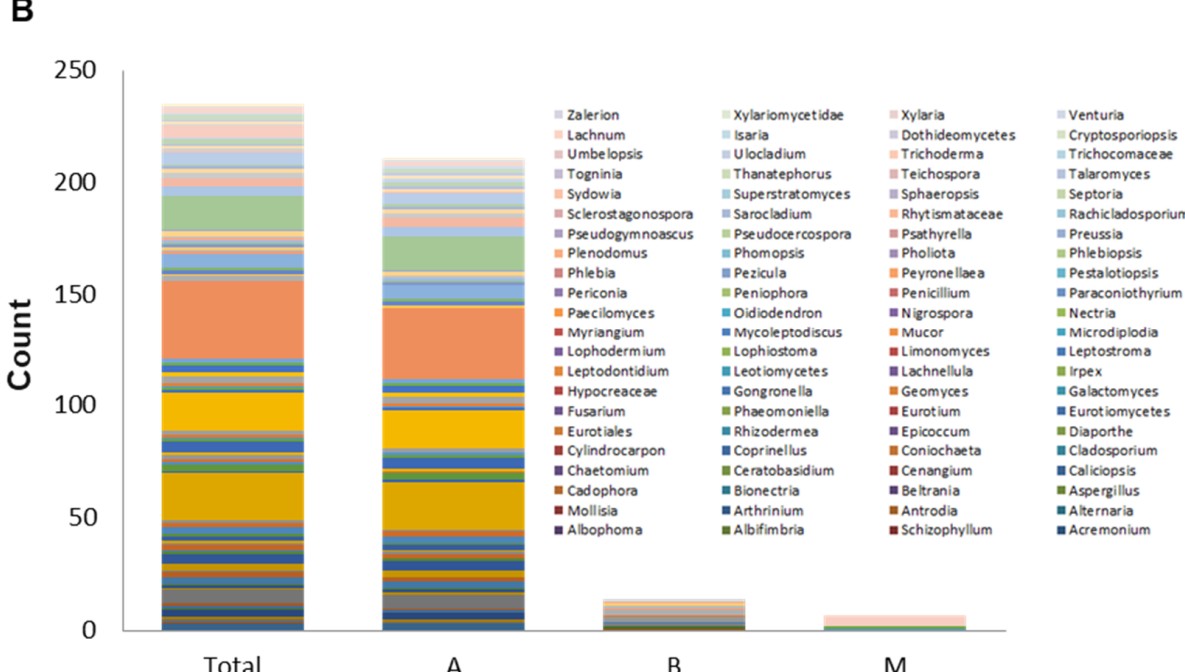

**Figure 2.** Isolation and identification of fungal endophytes. (**A**) Pie chart showing the percentage of isolated fungal endophytes belonging to the different phyla. (**B**) Bar plot showing the number of fungal species belonging to each of the 88 total genera recovered. Letters A, B and M represent the phyla Ascomycota, Basidiomycota and Mucoromycota, respectively.

### 3.2. Diversity of Fungal Endophytes among Sites

We found substantial variation in the fungal species richness and diversity across the sampling sites (Figure 1). Fungal diversity in a given site was different from those in other sites. For *P. densiflora*, site 2D showed the highest species richness (S) followed by sites 1D, 3D, 4D, and 5D. Site 1T revealed the highest S for *P. thumbergii*, followed by sites 2T, 3T, 5T, and 4T. The highest S for *P. koraiensis* was found in sites 3K, followed by 1K, 2K, and 4K. Site 1R for *P. rigida* showed the highest S followed by sites 1R, 4R, 2R, and 3R (Figure 3A). Rank wise analysis of S among sites (regardless of *Pinus* species) showed that site 1T has the richest species of fungal endophytes with 51 OTUs, and site 4T was observed to have the lowest species richness with 27 OTUs. On the other hand, Shannon diversity index (H') of the isolated fungal endophytes from *P. densiflora* revealed that site 2D has the highest H'

among other sampling sites. For *P.thumbergii*, site 2T showed the highest H′, site 1K for *P. koraiensis*, and site 1R for *P. rigida* (Figure 3B). For all the sampling sites, the highest H′ was observed in site 2D (H′ = 3.216), while the lowest index value was found in site 2K (H′ = 2.232).

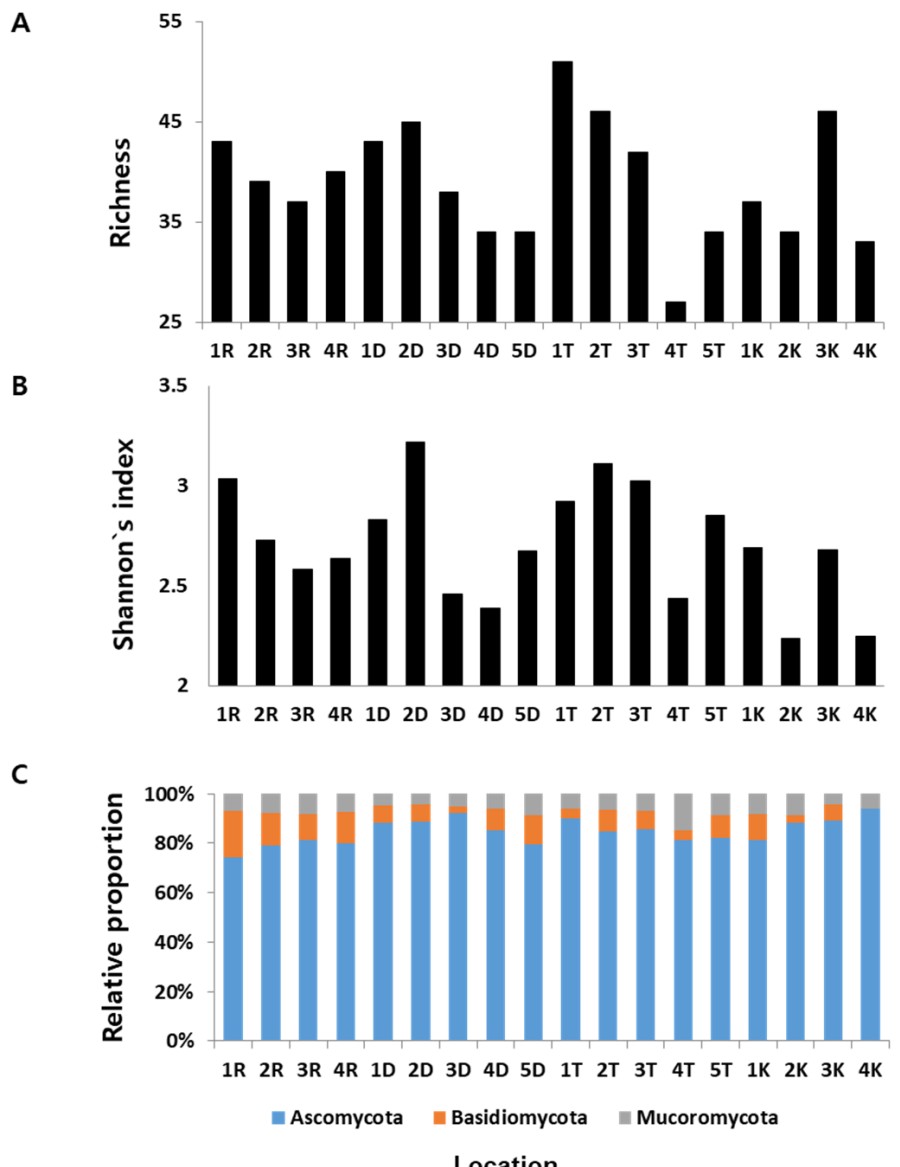

**Figure 3.** Distribution and diversity of fungal endophytes among sites. (**A**) Species richness across the sampling sites. (**B**) Shannon index (H′) across the sampling sites. (**C**) Relative proportion of fungal endophytes at phylum level across 18 sampling sites. Letters R, D, T, and K represent the *Pinus* species *rigida*, *densiflora*, *thunbergii,* and *koraiensis*, respectively.

We also analyzed the relative proportion of each phylum recovered from all sampling sites. As shown in Figure 3C, Ascomycota was found to be the most dominant phylum in all sites. The highest number of fungal endophytes belonging to Mucoromycota was recorded in site 4T. In the case of Basidiomycota, site 1R showed the highest number of fungal endophytes belonging to this phylum. No significant differences on the composition of fungal endophytic community were observed among different sampling sites (Supplementary Figures S1 and S2), although some specific site parameters (Ca and Mg ions and altitude) vary significantly (Supplementary Table S3).

### 3.3. Diversity of Fungal Endophytes within Tissues

We analyzed the diversity and distribution of fungal endophytes in the three different tissues (needle, stem, and root) of the four *Pinus* species. Significant differences were observed in the distribution of fungal endophytes among the different tissues (Figure 4). Pooled data from the three *Pinus* species revealed that the highest number of different fungal endophytes was in the root tissues (103 OTUs), followed by the needle (100 OTUs), and the stem (82 OTUs). Some endophytes were found only in one tissue, while 32.6% of all the endophytes (47 OTUs) were found to exist in all three tissues (Supplementary Table S4).

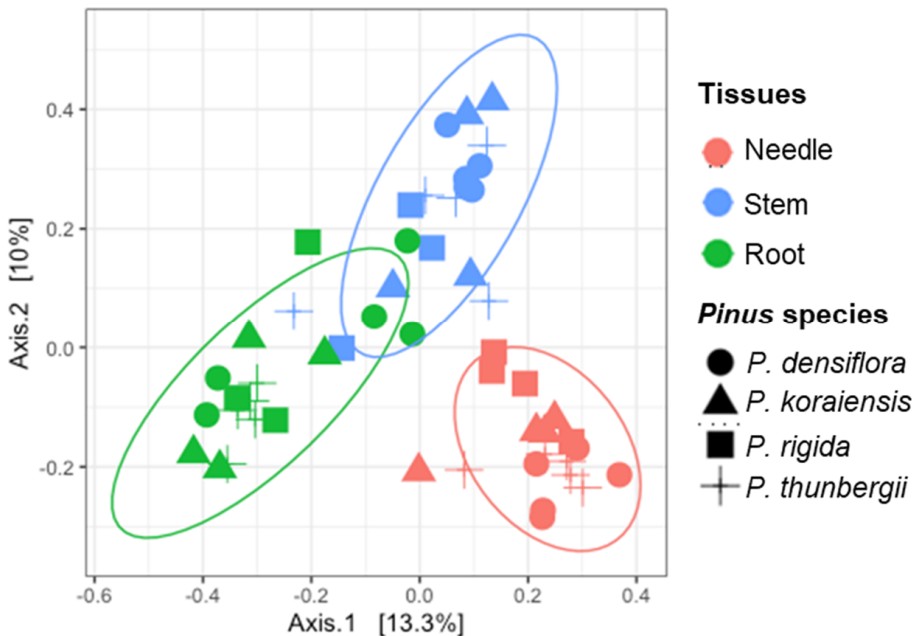

**Figure 4.** Principal coordinates analysis (PCoA) on Bray-Curtis distance showing the distribution of the fungal endophytes in the three different tissues (needle, stem, and root) of the four different *Pinus* species. Each color (pink, green, and blue) represents tissues (needle, stem, and root, respectively). Each shape (circle, triangle, square, and cross) means *Pinus* species (*P. densiflora*, *P. koraiensis*, *P. rigida*, and *P. thunbergii*, respectively).

The relative proportion of fungal endophytes in tissues (Figure 5A) showed that Ascomycota was the most abundant phylum, while Basidiomycota was found to be the least represented phylum in all tissues. In addition, the phylum Mucoromycota was more common in the stem and root tissues than in the needle tissues. Analysis of the species richness (S) in different tissues showed that the needle and root tissues had significantly higher S than in stem (Figure 5B). In terms of the diversity of fungal endophytes, no significant differences among different tissues were observed as measured by Shannon diversity index (Figure 5C). Pairwise comparison of the three tissues using count data of individual fungal endophytes showed that the stem and root tissues have a similar composition of fungal endophytes (Figure 5D). Interestingly, the fungal composition of needle tissue was more similar to root than to stem.

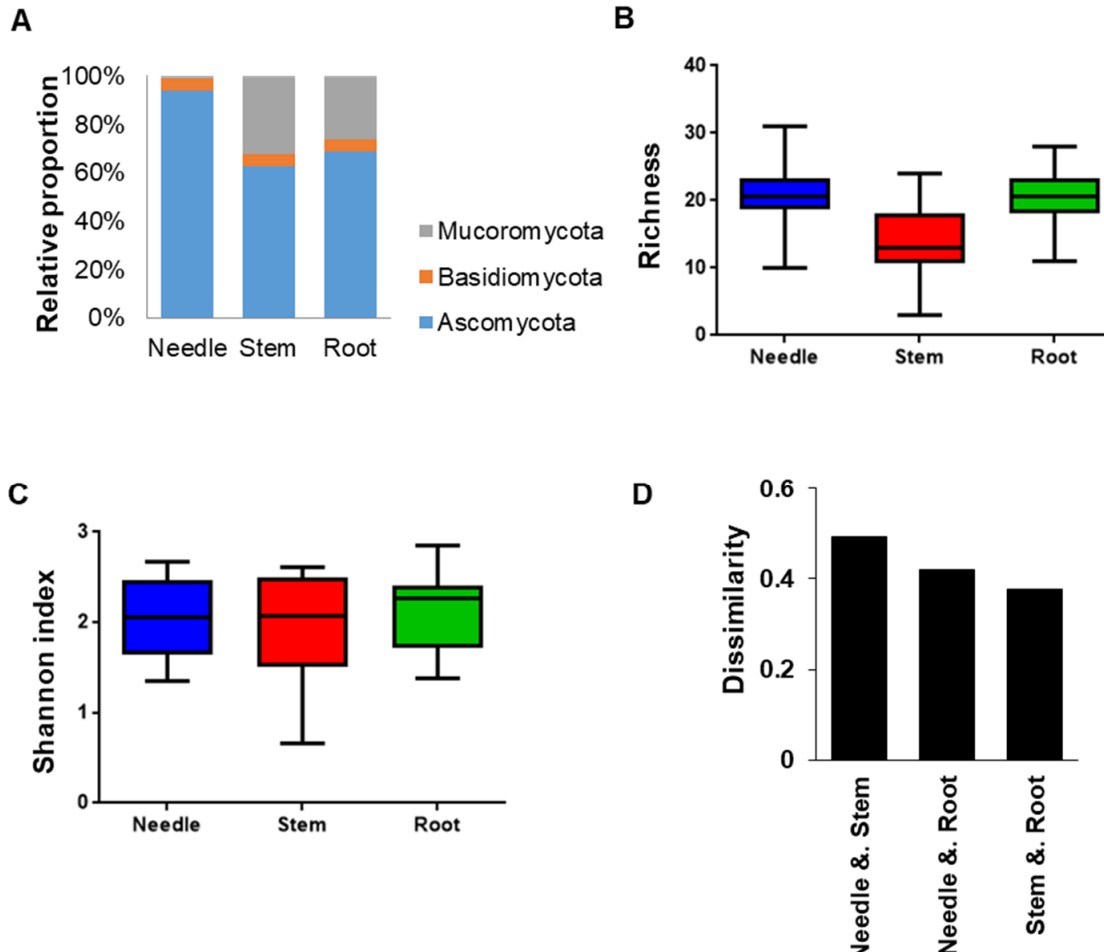

**Figure 5.** Distribution and diversity of fungal endophytes within different tissues of four *Pinus* species. (**A**) Relative proportion of endophytes in the three different tissues (needle, stem, and root). (**B**) Species richness (S) of fungal endophytes in three different tissues (needle, stem, and root). (**C**) Diversity index of the fungal endophytes in the three different tissues (needle, stem, and root) measured by Shannon diversity index. (**D**) Result of the dissimilarity analysis based on the composition of fungal endophytes from each tissue. Dissimilarity was calculated using Bray-Curtis dissimilarity measure.

### 3.4. Diversity of Fungal Endophytes within Pinus Species

We also examined the diversity and distribution of fungal endophytes in different *Pinus* species. Overall, the highest number of culturable fungal endophytes were isolated from *P. thunbergii* (1791), followed by *P. densiflora* (1469), *P. rigida* (1379), and *P. koraiensis* (1233). The culturable fungal endophytes represent 97, 90, 89, and 84 OTUs in *P. densiflora*, *P. thunbergii*, *P. rigida*, and *P. koraiensis*, respectively. Significant differences in the diversity of fungal endophytes were observed within *Pinus* species ($p$-value, 0.022; Table 1).

**Table 1.** Results of permutational multivariate analysis of variance (PERMANOVA) main test.

| Source | Degrees of Freedom | Sum of Squares | Mean Squares | *F*-Value | $R^2$ | Pr (>*F*) |
|---|---|---|---|---|---|---|
| Sites | 15 | 6.3426 | 0.42284 | 0.96556 | 0.28689 | 0.650 |
| Tissues | 2 | 2.9515 | 1.47573 | 3.77480 | 0.13350 | 0.001 ** |
| *Pinus* species | 3 | 1.7899 | 0.59663 | 1.40950 | 0.08096 | 0.022 * |

* significant $p$-value, $p < 0.05$, ** highly significant $p$-value, $p < 0.001$.

Among the four *Pinus* series, *P. thunbergii* showed the highest species richness followed by *P. densiflora* and *P. koraiensis*, and least in *P. rigida* (Figure 6A). In terms of diversity, as measured by Shannon diversity index, *P. thunbergii* consistently showed the highest diversity index followed by *P. densiflora*, *P. rigida* and *P. koraiensis* (Figure 6B).

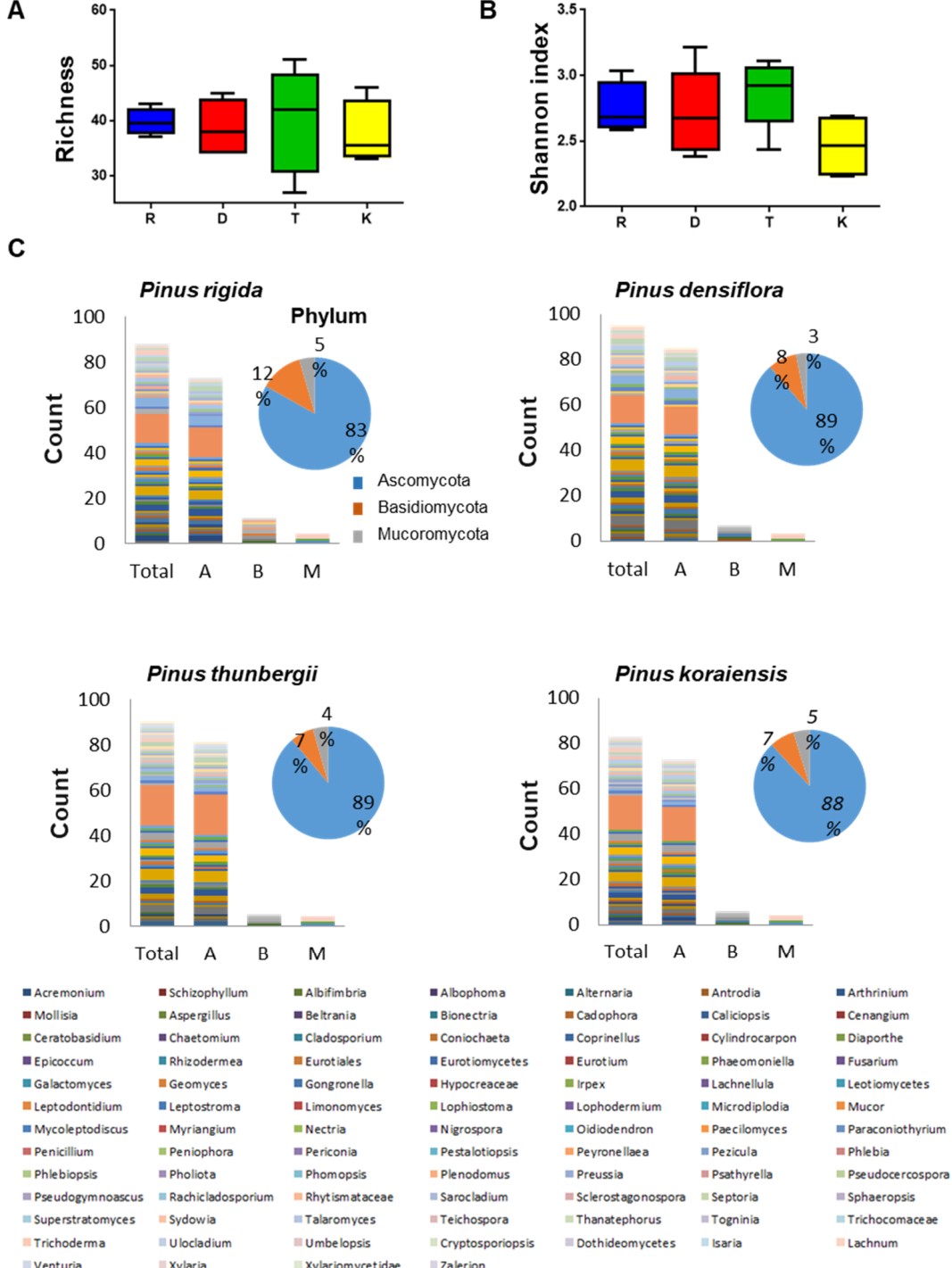

**Figure 6.** Distribution and diversity of fungal endophytes within four *Pinus* species. (**A**) Species richness in *P. rigida* (R), *P. densiflora* (D), *P. thunbergii* (T), and *P. koraiensis* (K). (**B**) Shannon index in *P. rigida* (R), *P. densiflora* (D), *P. thunbergii* (T), and *P. koraiensis* (K). (**C**) Bar plot showing the number of OTUs belonging to each of the 88 total genera recovered. The pie charts show the percentage of isolated fungal endophytes belonging to the different phyla. Letters A, B and M represent the phyla Ascomycota, Basidiomycota and Mucoromycota, respectively.

Analysis of the relative proportion of fungal endophytes (Figure 6C) in the four *Pinus* species showed that Ascomycota was the most abundant phylum and *Cladosporium cladosporioides* was the most abundant strain inhabiting the four *Pinus* species. Some fungal endophytes displayed host-plant specificity. For instance, *Ceratobasidium* sp., *Limonomyces roseipellis*, *Pholiota tremellos*, *Psathyrella* sp., *Pseudocercospora* sp., and *Ulocladium consortiale* were found exclusively in *P. rigida*. The following isolates, *Schizophyllum* sp, two fungal endophytes *Cylindrocarpon* sp., *Epicoccum nigrum*, *Eurotiales* sp., *Periconia* sp., and *Peyronellaea glomerata*, were isolated only in *P. densiflora*. In *P. thunbergii*, four distinct isolates were found: *Mycoleptodiscuc terrestris*, *Preussia funiculata*, *Togninia* sp., and *Venturia* sp. Two fungal endophytes, *Pseudocercospora* sp. and *Teichospora melanommoides* were only found to exist in *P. koraiensis*.

### 3.5. FUNGuild Analysis

FUNGuild was used to predict the nutritional and functional groups of the fungal communities in the needle, stem, and root tissues of the four *Pinus* species. The results showed that the fungal endophytes can be classified into eight trophic mode groups (Figure 7A), with pathotrophs, pathotroph–saprotroph symbiotrophs, and saprotrophs being the three major abundant trophic mode groups. In *P. rigida*, the needle and root tissues were dominated by saprotrophs, while the pathotroph–saprotroph symbiotroph groups dominated the stem tissue. The root tissue of *P. densiflora* was dominated by two trophic groups, namely the saprotrophs and the pathotroph–saprotroph symbiotroph group. The pathotroph group was abundantly found in the needle tissue. In *P. thunbergii*, pathotrophs were abundant in the needle and stem tissues. In addition, the stem tissue also housed a number of saprotrophic fungal endophytes, while the root tissue mainly composed of pathotroph–saprotroph symbiotroph. The saprotroph groups were highly abundant in the stem and root tissue of *P. koraiensis*, while the pathotroph–saprotroph symbiotroph group of fungal endophytes dominated the needle tissue. A proportion of symbiotrophs was also observed in the different tissues of the *Pinus* species, except in *P. rigida.* From the dominant trophic mode groups, we identified five putative fungal guilds (Figure 7B). The fungal guild groupings were represented by the animal pathogen–endophyte–lichen parasite–plant pathogen–wood saprotroph, plant pathogen, dung saprotroph–unidentified saprotroph–wood saprotroph, unidentified saprotroph, and including the unidentified guild. In general, these fungal guild groupings showed variable abundances in the three different tissues (needle, stem, and root) within a single *Pinus* species as well as between different *Pinus* species, suggesting the possible effect of host and tissue specificity on the relative abundances of the fungal guild groupings. Notably, the needle tissues of *P. densiflora*, *P. thunbergii*, and *P. koraiensis* were abundantly composed of fungal endophytes belonging to the plant pathogen fungal guild, while in *P. rigida*, unidentified saprotrophs dominated the needle tissue. The stem and root tissues of all *Pinus* species were dominated by fungal endophytes belonging to the unidentified saprotroph guild, except for the root tissue of *P. thunbergii*, where the animal pathogen–endophyte–lichen parasite–plant pathogen–wood saprotroph guild had a higher relative abundance.

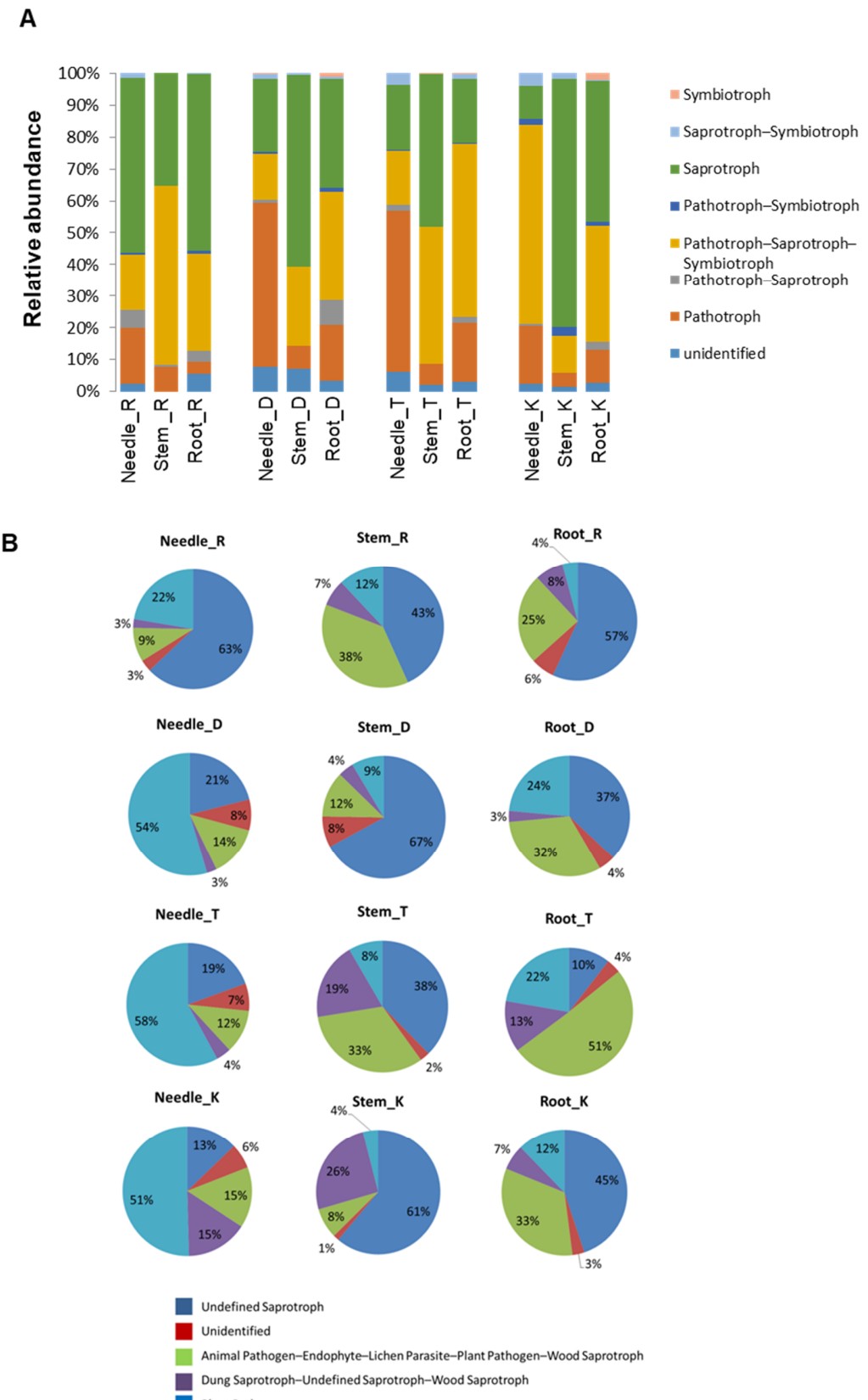

**Figure 7.** Relative abundance of predicted functional fungal endophytes. (**A**) Relative abundance of fungal trophic modes in needle, stem and root of the four *Pinus* species. (**B**) Relative abundance of fungal guilds in needle, stem and root of the four *Pinus* species. More data are supplied in Supplementary Figures S3 and S4.

## 4. Discussion

Endophytes are microbes such as fungi and bacteria that inhabit the tissue of their host plants in a symbiotic association. In general, the endophytes enter plant tissue through the root, stem, leaf, flower, and cotyledon. They can either live at the point of entry or spread throughout the plant tissues [45,46]. Fungal endophytes are ubiquitous in nature and are living virtually in all plants [47]. About 600,000 fungal endophyte species were thought to exist, and the majority of the endophytes are yet to be described [48].

In this study, we investigated the diversity of fungal endophytes in the four *Pinus* species in Korea. A total of 5872 culturable fungal endophytes were obtained from three different tissues (needle, stem, and root) of the four *Pinus* species obtained from 18 different sampling sites. The morphological characteristics and ITS or LSU rDNA sequencing analyses revealed that the fungal endophytes comprise 234 different fungal species with 144 OTUs belonging to 88 different genera. Of the 88 genera, 23 genera (*Alternaria*, *Aspergillus*, *Cadophora*, *Chaetomium*, *Coniochaeta*, *Epicoccum*, *Eurotiomycetes*, *Fusarium*, *Leotimycetes*, *Lophoderium*, *Microdiplodia*, *Nigrospora*, *Paraconiothyrium*, *Penicillium*, *Pestalotiopsis*, *Phomopsis*, *Preussia*, *Rachicladosporium*, *Sclerostagonospora*, *Trichoderma*, *Ulocladium*, *Dothideomycetes*, and *Xylaria*) were already reported to have been isolated as endophytes from pine trees in previous studies [13,21,22,31,35]. However, most of the remaining genera were not previously reported. This result substantially provides new additional genera to the existing literature of fungal endophytes isolated from pine trees. Moreover, our results further suggest that the composition of endophytic community from four *Pinus* species in Korea is highly diverse.

From the previous report, soil-borne fungi are usually more frequent and varied than those that inhabit the above-ground parts of plant [32]. It has been known that soil fungi from the genus *Umbelopsis* can produce polyunsaturated fatty acids, which are used in medicine, pharmacology, cosmetics, food industry, and agriculture [49]. In our study, *Umbelopsis* spp. were observed as one the fungal endophytes with the highest isolation frequency, dominating the root and stem tissues of the *Pinus* species. This result may further indicate that the soil fungi belonging to the genus *Umbelopsis* might be capable of systemic colonization, resulting in their existence in the different parts of the host plants. On the other hand, the species belonging to the genus *Lophodermium* are regarded as the most dominant fungal endophytes from *Pinus*, *Abies*, and *Picea* [22,31,50]. Interestingly, in our study, the species belonging to *Lophodermium* represent only 7.23%, together with some other common species of *Fusarium* (8.93%), and *Rhytismataceae* (6.38%). This difference might be due to the environmental differences as well as host species specificity of the fungal endophytes. The orders Rhytismatales, comprising the family *Rhytismataceae* and *Lophodermium* were also identified in this study. It has been reported that order Rhytismatales can cause foliar infections. Besides, within *Rhytismataceae* family, *Lophodermium pinastri* has been found to cause needle infections [48]. The presence of pathogenic fungal endophytes in our samples may indicate an opportunistic infection in pine trees. The endophytic lifestyle of these pathogenic fungi may serve as an entry point for the early stage of the infection process before transitioning to a pathogenic lifestyle.

Recently, many studies have been conducted to specifically reveal the composition of fungal endophyte communities in different tissues of plants [23,31,32]. Here, we revealed a varying composition of fungal endophytes in the three different tissues of the *Pinus* species. Some fugal endophytes in our study showed tissue specificity such as *Acremonium* spp., *Teichospora* spp., *Trichocomaceae* spp., *Isaria* spp., *Xylariomycetidae* spp., *Ceratobasidium* spp., and *Phlebia* spp., exclusively found in the needle tissue. Other fungal endophytes like *Epicoccum* spp., *Peyronellaea* spp., *Togninia* spp., *Ulocladium* spp., and *Venturia* spp. were specific to the stem tissue. Root tissue-specific fungal endophytes include *Cadophora* spp., *Cylindrocarpon* spp., *Ericoid* spp., *Eurotiales* spp., *Mycoleptodiscus* spp., *Periconia* spp., *Dothideomycetes* spp., *Xylariomycetidae* spp., *Pholiota* spp., and *Psathyrella* spp. Furthermore, most of these fungal endophytes were isolated for the first time from each tissue except *Epicoccum* spp., *Ulocladium* spp., *Cadophora* spp., and *Dothideomycetes* spp. The tissue specificity of some fungal

endophytes demonstrated here, might suggest that they are well adapted to survive within the specific chemistry or texture of a particular host tissue [51]. Characterization of these tissue-specific fungal endophytes should be performed in future studies to understand their nature of tissue colonization as well as their roles as endophytes.

Fungal endophytes are affected by environmental variations such as climatic condition, geographic situation, edaphic factors, and host factors like plant genotypes [25,51–53]. Surprisingly, of all the environmental factors evaluated, no significant effects on the fungal community composition were observed. These results further suggest that the composition of fungal community in *Pinus* species in Korea is not greatly affected by various environmental factors and might be shaped independently regardless of the varying environmental conditions. In our study, Ascomycota showed to be the most commonly occurring phylum associated with the four *Pinus* species. This result corroborates with the previous studies conducted in *Pinus* species [21,35]. Moreover, *Cladosporium* spp. are the dominant species found in the three *Pinus* species (R, D, T), while *Umbelopsis* spp. appeared to be the dominant fungal species in *P. koraiensis* (K). It has been known that some fungal endophytes exhibit host-specificity. Furthermore, assessing the diversity of fungal endophytes among the four *Pinus* species demonstrated that *P. thunbergii* has the highest species richness followed by *P. koraiensis*. *P. thunbergii* and *P. koraiensis* have been reported to grow and tolerate adverse environments [13,18]. *P. thunbergii* can grow under drought stress and can tolerate high salinity in the coastal forest of Korea [13]. *P. koraiensis* on the other hand can thrive under acidic conditions [16]. The tissue microenvironment of *P. thunbergii* and *P. koraiensis* could possibly provide favorable conditions for a variety of fungi to thrive as endophytes. In turn, potential beneficial and growth promoting endophytes can boost the growth and resistance of these pine species to withstand hostile environmental conditions.

Our study applied FUNGuild to predict and to better understand the functions of fungal endophytes. It has been reported that the higher proportion of saprotrophs and pathotrophs compared to symbiotrophs is associated to the emergence of plant diseases [20]. In our study, high abundance of saprotrophs and pathotrophs was identified, suggesting a potential element to the decline of *Pinus* species in Korea. Saprotrophs are the primary decomposers of dead or aged plants and play an important role in decomposing organic matter and nutrient cycling [54]. Interestingly, in *P. densiflora* and *P. koraiensis*, saprotrophs had significantly higher percentage in stem compared to other tissues. This can be explained by the high proportion of known saprotrophs *Umbelopsis isabellina* and *Mucor circinelloides* in the stem of *P. densiflora* and *P. koraiensis*. *Umbelopsis isabellina* can produce not only protease as wood decaying agent but also can produce fatty acids that aid in the decaying process [49,55]. *Mucor circinelloides* is used for commercial fermentations and is predominantly saprotrophic [56]. All *Pinus* species, except *P. rigida* showed the highest fungal endophytes belonging to plant pathogen guild in the needle compared to other tissues. This result suggests that most of the fungi having tendencies to cause diseases were localized in the needle tissues. We further observed that the most abundant fungal strain *Septoria* sp. in needle has been long known as a leaf pathogen in plants [57]. Thus, we could speculate that fungal pathogens of pine trees break through the needle tissue and eventually proceed with the infection process.

## 5. Conclusions

In summary, we revealed the composition and diversity of fungal endophytes in the three different tissues (needle, stem and root) of four *Pinus* species in Korea. We also showed that some fungal endophytes have preferences in terms of the host and tissues they colonize. Environmental conditions have no significant impact on the overall composition of fungal endophytes in three different tissues of the four *Pinus* species. We also found that the *Pinus* species are more prone to needle diseases as demonstrated by the high abundance of plant pathogenic fungi in needle tissue. Moreover, the high proportion of saprotrophs and pathotrophs found in the *Pinus* species, which implicates a possible emergence of plant disease could be one of the probable causes of the deterioration of *Pinus* species in

Korea. Overall, these findings are crucial for the better understanding of the diversity and communities of fungal endophytes associated with the four *Pinus* species in Korea and their possible implication on the status of these *Pinus* species. Our findings also provide a starting reference for the exploitation of these fungal endophytes for practical applications.

**Supplementary Materials:** The following are available online at https://www.mdpi.com/1999-4 907/12/3/302/s1. Table S1: Characteristics and conditions of 18 sampling sites in Korea. Table S2: The closest relatives of fungal endophytes isolated from pine trees based on the sequence analyses of the nuclear ribosomal internal transcribed spacer region (rDNA ITS) and nuclear large subunit ribosomal DNA (LSU rDNA). Table S3. Result of the permutational multivariate analysis of variance (PERMANOVA) between different environmental factors found in all sampling sites. Table S4. Summary of 144 OTUs isolated from three tissues. Figure S1: Scatter plots showing relationships between species richness and the factors of sampling sites. Figure S2: Scatter plots showing relationships between Shannon's index and the factors of sampling sites. Figure S3: Relative abundance of Trophic mode in *Pinus* species (A) *P. rigida*, (B) *P. densiflora*, (C) *P. thunbergii*, and (D) *P. koraiensis*. Blue, green and pink bars mean root, stem, and needle, respectively. The data were analyzed by employing the analysis of variance and Tukey's test ($p < 0.05$) using R version 3.6. Significance codes: 0 \*\*\*, 0.001 \*\*, 0.01 \*. Figure S4: Relative abundance of Fungal Guild in *Pinus* species (A) *P. rigida*, (B) *P. densiflora*, (C) *P. thunbergii*, and (D) *P. koraiensis*. Blue, green and pink bars mean root, stem, and needle respectively. The data were analyzed by employing the analysis of variance and Tukey's test (*p < 0.05*) using R version 3.6. Significance codes: 0 \*\*\*, 0.001 \*\*, 0.01 \*.

**Author Contributions:** S.O.R. performed most of the experiments; S.O.R., M.R. and J.J. analyzed the data; S.O.R., J.A.V.M., S.-C.P. and H.B. wrote the article with contribution of all the authors. H.B. supervised the experiments. All authors have read and agreed to the published version of the manuscript.

**Funding:** This research was supported by Yeungnam University Research Grant 218A380135.

**Institutional Review Board Statement:** Not applicable.

**Informed Consent Statement:** Not applicable.

**Data Availability Statement:** The data used in this study are available on request from the corresponding author.

**Conflicts of Interest:** The authors declare no conflict of interest.

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
