# Peer review of "Diversity and Communities of Fungal Endophytes from Four Pinus Species in Korea"

_forests, doi:10.3390/f12030302_

Round 1

Reviewer 1 Report

The study is aimed at diversity of culturable fungal endophytes in three types of tissues of four pine species in Korea. Multiple sites were surveyed and tissue samples yielded huge number of fungal strains. These were identified based on their ITS rDNA and characterised according to FUNGuild tool. Numerous statistical analyses and also simple phylogenies were performed to show how fungal diversity, abundance and fungal community structure depend on specific site parameters and graphical elements are clear and well arranged. The number of fungal species isolated and distinguished, 234 is amazing. Unfortunately, the text is very difficult to follow, the presentation of results is so detailed that it cannot be understand and the text suffers from numerous inconsistencies, inaccuracies, misunderstandings and obvious errors resulting from limited knowledge of fungi, their taxonomy and ecology. Details are provided bellow:

Introduction

This chapter is too long, too general and present straightforward statements about fungal ecology that are however not supported by given literature references or are even erroneous. In general, vast majority of references is completely inappropriate, like if they were selected randomly. As an example, four out of five first references do not make sense:

  1. 36 – the very general definition of endophytes is accompanied by a reference to a case study of one fungal endophyte in seed of Phragmites
  2. 38 – the completely erroneous statement about endophytes providing nutrition to plants (how?) is accompanied by reference to a study of pine needle endophyte diversity
  3. 40 – contribution of endophytes to plant nutrition by synthesis of various hormones is followed by a reference to a recent paper of senior author with not a single word about plant hormones
  4. 44 – the statement that “many studies” demonstrated biocontrol effect of endophytic fungi is followed by just one(!) reference, review of biocontrol potential of endophytic bacteria co-authored by the senior author

On the other hand, statement about “only a few examples … in tolerance against pathogens” (l. 49) ignores numerous studies on Phialocephala scopiformis producing rugulosin.

M & M

In general, clearly written collection design and isolation contain only couple of inaccuracies, such as rinsing with tap water “to remove the soil” – probably only roots, but mentioned in general for “samples”. Molecular identification is “spoiled” by nonsense references 32 and 33 referring to bacterial communities and transposons! Data analysis is chaotic, first sentence doesn’t make sense at all. “Various conditions” tested are mentioned very briefly without explanation – how were e.g. organic matter, silicic acid and three selected ions evaluated, where (soil, nutrients in plant tissues?) and why particularly these factors were selected?

Results

Presentation of results is very poor from the beginning. It would be much better to differentiate between number of strains isolated and number of OTUs recognized, because the use of “fungal endophytes” and “fungal isolates”, doesn’t make sense. The huge Table 1 is completely redundant and also presentation of genera in Results and Discussion. Genus is not a biological entity, just a taxonomic construct, and pooling together several species (actually unknown number for reader) and assign a putative function to them is erroneous (especially at genera having >500 species, such as Cladosporium, Pholiota, Penicillium, etc.). Moreover, “Agaricaceae, Eurotiomycetes”, etc. refer to higher taxonomical categories and reflect missing referencing data in GenBank. Also the two strains identified as “Fungal sp.” and belonging to “Unknown” phylum cannot be treated this way. According to the simple phylogenies S2 A and B, they clustered among other ascomycetes. Btw., the phylogenies are completely erroneous and full of artefacts. ITS cannot be used to build a Fungi-wide phylogeny and, as a result, both ascomycetes and basidiomycetes are mixed up. Moreover, I do not see a reason to make three phylogenies depending on the isolate source; their position should be still the same no matter if they come from needle, root or stem. The table S2 makes better sense, but still contains errors (Sydowia is an ascomycete) and improper names (“Ericoid mycorrhizal sp.” is not a taxonomical category).

Calculation of diversity indices and various comparisons is simply chaotic because of too general and repeated statements. It should be clarified, what author mean with “richest composition”, “composition of fungal endophytic community”, “highest number of different fungal endophytes”, “count data of individual isolates”, “composition of fungal endophytes”, “different endophytes”, etc. Notably, the “isolates belonging to Unknown …” mentioned three times throughout the Results should be omitted, there is no such a taxonomical category! Sorry to say, but the FUNGuild part with frequencies of particular guilds in various pines and tissues is impossible to follow (what are “most overrepresented fungal guild groups”?) and data should better be presented as pie charts or box plots, such as in Fig. 4.

Discussion

First two paragraphs just repeat results followed by irrelevant references, but, interestingly, another study aimed at fungal endophytes in pine needles in Korea: Kor. J. Mycol. 37(2) : 130-133 (2009) is not mentioned in the manuscript at all!. The largest part between l. 352 and 385 is very problematic and probably represents the poorest part of the whole manuscript. Particular fungal genera are combined with (randomly?) selected ecology of some “representative” species and followed by (randomly?) selected references (probably Google scholar search with particular fungal genus). I cannot go through them one by one, but these conclusions just cannot be drawn this way! Fig. 6 suffers from not defining what are “respective reference fungal endophytes” and how were “important fungal endophytes” selected and its informative value is null. Table 4 that concentrates and repeats  “conclusions” from the problematic part between l. 352 and 385 should be completely omitted; by not defining the “Benefit/drawbacks” (for host? ecosystem? humans?), one may just guess, what should this table say. Towards the end, the discussion is easier to understand, but authors mostly repeat results and already well known facts (ascomycetes are the most common endophytes, medium affects isolation success) and only the very end brings an interesting and original deduction, however very speculative (tissue microenvironment was not analysed in this study).

Reviewer 2 Report

This has been very huge study and deserves publication. However it need to be improved and most important is that the authors check all of the references as they are not correctly cited. Similarly, more references are needed in the discussion. There is a lot done in endophytes in these tissues. Main studies seem to be missing. Concentrate also in the discussion only to the endophytes from needles (and compare to other studies), stem and roots. I don't like the idea of comparing all the endophytes isolated from trees. As already Rodriquez et al. 2009 in: Fungal endophytes: diversity and functional roles, stated that root endophytes are strictly only in roots etc. These cannot be compared. Concentrate on the tissues between different species. 

Review of  Diversity and communities of fungal endophytes from four Pinus species in Korea

Abstract: A single = can you remove this? I don’t see how this is needed in the sentence

In total, 234 different fungal isolates were classified….

But FIRST you classified all of them based on morphology. Then you used ITS/28S to identify the species. Mention first the morphological classification and then the identification but don’t use the word classified in that content (molecular identification)

 some fungal endophytes showed host and tissue specificity. (not surprise as it is known that under and below ground fungi are different, see review Fungal endophytes: diversity and functional roles)

Introduction

You need to check ALL REFERENCES and search for better references throughout the article:

e.g. reference no 1, you can find better ones e.g. Petrini 1990 who define what endophyte is

Also for the reference 2, (A community of unknown, endophytic fungi in western white pine. )  this study does not state the fact you are claiming in your text. If you say that endophytes provide as nitrogen, phosphorus, and irons to the host, I want to check the references and found out studies that prove this.

Reference 3:

 Moreover, they also exhibit the ability to synthesize plant growth hormones, such as gibberellins, indole acetic acid, and auxin in the reference list is the study: Fungal endophytes inhabiting mountain-cultivated ginseng (Panax ginseng Meyer): Diversity and biocontrol activity against ginseng pathogens. And this study does not say that endophytes can synthesize these plant hormones

For the statement where you have reference 39 better would be:

Jumpponen & Trappe New Phytol. (1998), 140, 295–310, Dark Septate Endophytes: Facultative Review

Scientific names with Italics

Line 100:

This study demonstrated the specific relationship between salt-tolerant fungal endophytes with P. thunbergii living closer to the coast. Thus, it can be noted from these previous studies that the diversity and community composition of fungal endophytes in Pinus sp. can be affected by several factors, for example tree ages, seasonal changes [20,22,23] and other environmental factors [26].

Should be in discussion

Material and methods:

What kind of root samples was collected? Fine roots with visible mycorrhiza? Similarly, how you define stem sample? Did the stem have bark, phloem and sapwood? Please give more information.

The primers need to have the correct references!

As we know the species composition is different between tissues can you calculate the relative abundance by using all isolates from all tissues. I would suggest to calculate it for each tissue

Do you mean altitude or latitude?

Results

Make clearer table 1 for the results. There is no species listed only the genus. Can you have species? It can be only TOP 20 to 50 species. OR TOP 10 to each plant tissue. And then provide the proportions for each tree species.

Discussion:

No Figures or tables in discussion

Papers that you can add to the study to confirm your results:

Latitude: see Terhonen et al. 2011 The effect of latitude, season and needle-age on the mycota of Scots pine (Pinus sylvestris) in Finland. Silva Fennica 45: 301–317.

Nguyen et al:

https://doi.org/10.1016/j.funeco.2016.07.003

Endophytes between different Pinus species:

Oliva et al 2020: https://doi.org/10.1111/1365-2435.13692

Round 2

Reviewer 1 Report

I appreciate the opportunity to read the improved version and I see some improvement compared to the previous version, indeed. This version almost follows standard criteria for scientific outputs , but still, numerous changes must be made so that it may represent a valuable contribution to the study of diversity of fungal endophytes worldwide.

Introduction

Most of the irrelevant references were omitted, but still, the scope of the chapter is excessively wide. Study of fungal endophytes in pine tissues does not have to be introduced with definition of fungal and BACTERIAL endophytes, references to studies on arbuscular mycorrhiza (absent in pines), maize and tropical plants. Please, focus on conifers (ev. trees) only.

M & M

The methods of the research are better explained with the exception of the (most important) parts, chapters 2.2-2.4. Chapter 2.2 ends with description of “identification of morphospecies” and chapter 2.3 begins with information that ITS and LSU were sequenced from all fungal endophytes. Was DNA extracted from ALL 5872 isolates, indeed? Considering the data analysis, I have an idea, how the abundances were calculated, but statement that “percentage of the number of all fungal isolates belonging to particular total isolates” is really confusing. Here and also on other places (e.g. “Taxonomic binning”), English must be improved so that the text is possible to understand. Fig. 2 is redundant, since the design is not original.

Results

I suggest that you rewrite the 3.1 chapter, so that it is clear that you obtained 5 872 cultures (=isolates) and they were sorted into 234 morphotypes (morphospecies) that were classified into 144 OTUs based on sequence identities. If this my understanding is incorrect, the clarifying of the text is even more necessary! Pie chart 3A is redundant, only repeats data from the text. Bar plot or the legend needs improving too; the text says that 144 OTUs were defined, so how is it possible that the “the number of OTUs” reaches almost 250.

Chapter 3.2 is problematic from several reasons – the text mostly repeats information from Fig. 4 and contains erroneously written statements. Not that “Mucoromycota was found to be the most abundant phylum in sites 4T and 5D”, but “Highest abundance of Mucoromycota was recorded in sites 4T and 5D”. The Permanova analysis and Table 1 is confusing and thereof deduced significant effect on the composition of fungal endophytes is not clear to me. In fact, the whole ending of the chapter is confusing in that fungal abundances are not connected with particular env. factors at the localities, that are analysed separately.

Chapter 3.3 is clearly written, but contains conflicting statements: “Overall, significant differences were observed among different tissues (Fig. 5).” vs. “In terms of the diversity of fungal endophytes, no significant differences among different tissues were observed”. This must be clarified in the next version.

Chapter 4.4 follows the same structure as previous chapters with relatively detailed descriptions of summary data with minor inaccuracies. E.g. abbrev. “spp.” means more than one species. Therefore more than “Two fungal endophytes, Pseudocercospora spp. and Teichospora sp. were only found to exist in P. koraiensis.” This statement needs also polishing, potentially, these endophytes may “exist” also somewhere else.

Chapter 3.4 is in a better shape.

Discussion

This chapter is still problematic with trivial statements (“Saprotrophs are the primary decomposers of dead or aged plants“),  discussion of GENERA that were recorded for the first time, already known from pines and their potential effect, potential use by humans and complicated deductions from obtained data. E.g. why is the use of fatty acids produced from Umbelopsis mentioned? Why is it notable that Umbelopsis is capable of vertical transfer when ALL endophytes colonize plants by vertical transfer??? Do I understood it correctly that Umbelopsis isabelina … can produce fatty acids that aid in the decaying process.” and therefore represents a “potential element to the decline of Pinus species in Korea?” The part with Lophodermium and Rhytismataceae and then of Rhytismatales comprising Rhytismataceae and Lophodermium is confusing.

Conclusion

Trivial and oversimplified statements present in this chapter are not based on obtained results, but on textbook facts – fungi have preferences for tissues, are affected by env. factors and needles have the highest abundance of fungal pathogens compared to other parts.

Author Response

Responses to Reviewer 1

Comments and Suggestions for Authors

I appreciate the opportunity to read the improved version and I see some improvement compared to the previous version, indeed. This version almost follows standard criteria for scientific outputs, but still, numerous changes must be made so that it may represent a valuable contribution to the study of diversity of fungal endophytes worldwide.

Introduction

Most of the irrelevant references were omitted, but still, the scope of the chapter is excessively wide. Study of fungal endophytes in pine tissues does not have to be introduced with definition of fungal and BACTERIAL endophytes, references to studies on arbuscular mycorrhiza (absent in pines), maize and tropical plants. Please, focus on conifers (ev. trees) only.

Response:

We intentionally wrote the introduction part in a fashion that it covers the general concepts of endophytes down to the specific topics (pine trees and fungal endophytes) in a hope to convey an understandable background of the study. For that reason, some studies not pertaining to pine trees and their associated fungal endophytes, like the ones you have mentioned were included in the text. To further highlight the specific scope of our study, we added reviews of literature focusing on fungal endophytes in conifers, as suggested. Please see quoted text.

Lines 60-82:

Previous studies have reported the isolation of fungal endophytes from many different tree species including pine trees [13,20–35]. However, most studies conducted in pine trees focused on the distribution and composition of fungal endophytes from needles [23,26,33] and found the phylum Ascomycota as the most abundant fungal endophytes from needle tissue [35]. However, fungal endophytes from pine tree in other tissues like roots and stems are still poorly explored and characterized. Thus, the distribution and composition of the fungal endophyte community associated with pine tree species in different parts (needle, stem and root) from different regions are extremely important to reveal the identity of unexplored endophytes. Several studies have demonstrated the effect of environmental and host factors in the composition of fungal endophytes inhabiting Pinus species. For instance, the study of Wang et al. 2007 [25] revealed the presence of Phomopsis archeri, Alternaria alternate, and Pestalotiopsis besseyi in stem, and Leptostroma sp. and P. lingam in needle tissues as the frequently isolated fungal endophytes of P. tabulaeformis, a major tree in northern Chinese forests. Moreover, it has been shown also that varying seasons and types of tissues have a significant impact on the number of isolated fungal endophytes [25]. In Spain, various pathogenic fungal endophytes such as Davidiella tassiana, Ulocladium sp., Epicoccum nigrum, Phoma herbarum, Pleospora herbarum, Aureobasidium pullulans and Phaeomoniella sp. were isolated in one of the most abundant conifers, P. halepensis [35]. The data on the isolated fungal endophytes have been correlated to the decline of P. halepensis and further provided an understanding on endophyte-host interactions. On the other hand, diverse salt-tolerant fungal endophytes, many of them were Penicillium sp., were isolated from the black pine, P. thunbergii in Korea [30].

M & M

The methods of the research are better explained with the exception of the (most important) parts, chapters 2.2-2.4. Chapter 2.2 ends with description of “identification of morphospecies” and chapter 2.3 begins with information that ITS and LSU were sequenced from all fungal endophytes. Was DNA extracted from ALL 5872 isolates, indeed?

Response:

Thank you for pointing out this matter. Not all the 5,872 isolates were subjected to molecular identification based on ITS/LSU. These 5,872 isolates were initially identified/grouped based on morphological characteristics resulting in the identification of 234 morpho-species. These 234 morpho-species of fungal endophytes were then subjected to molecular identification. In the revised manuscript, we have included a statement in the method section to clearly elaborate this step.   

Lines 133-137:

The fungal isolates were initially identified as morpho-species. For the identification of morpho-species, fungal isolates from pure cultures were characterized based on mycelium color, structure, conidia type, and conidiophore morphology (size, color, shape, ornamentation, etc.). The identified morpho-species were then subjected to molecular identification.

Considering the data analysis, I have an idea, how the abundances were calculated, but statement that “percentage of the number of all fungal isolates belonging to particular total isolates” is really confusing.

Response:

We have changed the statement as follows:

Lines 151-153:

The relative abundance/proportion was estimated as a percentage of the number of specific fungal isolates, OTUs or phylum divided by its total number.

Here and also on other places (e.g. “Taxonomic binning”), English must be improved so that the text is possible to understand. Fig. 2 is redundant, since the design is not original.

Response:

The text has been changed as follows:

Lines 177-179:

Further identification of the morpho-species based on rDNA ITS or LSU rDNA revealed 88 genera with a total of 144 operational taxonomic units (OTUs).

Fig. 2 has been removed.

Results

I suggest that you rewrite the 3.1 chapter, so that it is clear that you obtained 5,872 cultures (=isolates) and they were sorted into 234 morphotypes (morphospecies) that were classified into 144 OTUs based on sequence identities. If this my understanding is incorrect, the clarifying of the text is even more necessary!

Response:

You have certainly understood it correctly. We apologize for the misleading information. As recommended, this section of the result has been reconstructed to convey a more vivid presentation of the results. We changed the result as follows:

Lines 171-185:

We isolated a total of 5,872 culturable fungal endophytes (isolates) from three different tissues of four Pinus species across 18 different sampling sites in Korea. Root tissue presented the largest number of fungal endophytes (n = 2,528) followed by the needle (n = 2,381), and the stem tissues (n = 963). The total number of isolated culturable fungal endophytes varied depending on Pinus species: P. thunbergii (n = 1,791), P. densiflora (n = 1,469), P. rigida (n = 1,379), and P. koraiensis (n = 1,233). From the 5,872 fungal isolates, 234 morpho-species were identified. Further identification of the morpho-species based on rDNA ITS or LSU rDNA revealed 88 genera with a total of 144 operational taxonomic units (OTUs). The sequences were deposited in GenBank (Supplementary Table S2). The fungal endophytes belonged to three known phyla, namely Ascomycota, Basidiomycota, Mucoromycota, representing 91.06%, 5.95% and 2.97%, respectively (Fig.2A). Most of the identified species belonged to Penicillum (14.89%) of Ascomycota, and Umbelopsis (2.12%) of Mucoromycota (Fig.2B). The most frequently occurring phylum Ascomycota, included 127 OTUs. Basidiomycota group was represented by 13 different OTUs. Mucoromycota was represented by 4 OTUs.

Pie chart 3A is redundant, only repeats data from the text. Bar plot or the legend needs improving too; the text says that 144 OTUs were defined, so how is it possible that the “the number of OTUs” reaches almost 250.

Response:

We opted to indicate a pie chart as a representation of what has been written in the text to give a better picture of the results. For the bar plot, the caption has been miswritten, it should be fungal species not OTUs.

Figure 2. Isolation and identification of fungal endophytes. (A) Pie chart showing the percentage of isolated fungal endophytes belonging to the different phyla. (B) Bar plot showing the number of fungal species belonging to each of the 88 total genera recovered. Letters A, B and M represented the phylum Ascomycota, Basidiomycota and Mucoromycota respectively.

Chapter 3.2 is problematic from several reasons – the text mostly repeats information from Fig. 4 and contains erroneously written statements. Not that “Mucoromycota was found to be the most abundant phylum in sites 4T and 5D”, but “Highest abundance of Mucoromycota was recorded in sites 4T and 5D”.

Response:

We structured this section of the results in a way that it will explicitly explain what have been shown in the figures presented. We believe that in this manner, the results can be easily followed. We admit that some statements were confusing, and thus changes have been made ensuring that your concerns have been addressed as follows:

Lines 211-212:

The highest number of fungal endophytes belonging to Mucoromycota was recorded in site 4T.

The Permanova analysis and Table 1 is confusing and thereof deduced significant effect on the composition of fungal endophytes is not clear to me. In fact, the whole ending of the chapter is confusing in that fungal abundances are not connected with particular env. factors at the localities, that are analysed separately.

Response:

Thank you for pointing this matter. We have run another analysis of our data and found that the composition of fungal endophytes in the different sampling sites does not significantly vary. Although when we analyzed some site parameters, significant differences in Ca and Mg ions content together with altitude were observed among the sampling sites. These data suggest that environmental factors have no significant impact on the composition of fungal endophytes.

The text has been changed and the PERMANOVA table has been moved in the supplementary data as follows:

Lines 213-216:

No significant differences on the composition of fungal endophytic community were observed among different sampling sites (Supplementary Fig. S1 and S2), although some specific site parameters (Ca and Mg ions and altitude) vary significantly (Supplementary Table 3).

Chapter 3.3 is clearly written, but contains conflicting statements: “Overall, significant differences were observed among different tissues (Fig. 5).” vs. “In terms of the diversity of fungal endophytes, no significant differences among different tissues were observed”. This must be clarified in the next version.

Response:

We actually meant the “overall” distribution of fungal endophytes among the different tissues as determined by PCoA analysis. We apologize for the confusion brought by our statement. We have changed the sentence to clearly convey what is meant as follows:

Lines 224-226:

Significant differences were observed in the distribution of fungal endophytes among the different tissues (Fig. 4).  

Lines 236-237:

In terms of the diversity of fungal endophytes, no significant differences among different tissues were observed as measured by Shannon diversity index (Fig. 5C).

Chapter 3.4 follows the same structure as previous chapters with relatively detailed descriptions of summary data with minor inaccuracies. E.g. abbrev. “spp.” means more than one species. Therefore more than “Two fungal endophytes, Pseudocercospora spp. and Teichospora sp. were only found to exist in P. koraiensis.” This statement needs also polishing, potentially, these endophytes may “exist” also somewhere else.

Response:

We have change the binomial nomenclature of the fungal endophytes mentioned whenever available. We have also carefully checked that the mentioned fungal endophytes were indeed, exhibited host-specificity as per our data only. We have changed the sentences as follows:

Lines 275-282:

For instance, Ceratobasidium sp., Limonomyces roseipellis, Pholiota tremellos, Psathyrella sp., Pseudocercospora sp., and Ulocladium consortiale were found exclusively in P. rigida. The following isolates: Schizophyllum sp, two fungal endophytes Cylindrocarpon sp., Epicoccum nigrum, Eurotiales sp., Periconia sp., and Peyronellaea glomerata were isolated only in P. densiflora. In P. thunbergii, four distinct isolates were found: Mycoleptodiscuc terrestris, Preussia funiculata, Togninia sp., and Venturia sp. Two fungal endophytes, Pseudocercospora sp. and Teichospora melanommoides were only found to exist in P. koraiensis.

Chapter 3.4 is in a better shape.

Discussion

This chapter is still problematic with trivial statements (“Saprotrophs are the primary decomposers of dead or aged plants“),  discussion of GENERA that were recorded for the first time, already known from pines and their potential effect, potential use by humans and complicated deductions from obtained data. E.g. why is the use of fatty acids produced from Umbelopsis mentioned?

Response:

We mentioned some features of Umbelopsis, as this fungal endophyte represented the most frequently isolated fungal endophytes here in our study.

Why is it notable that Umbelopsis is capable of vertical transfer when ALL endophytes colonize plants by vertical transfer???

Response:

Umbelopsis is soil-borne fungi. In our study, Umbelopsis species were observed to dominate the root and stem. So we therefore postulated that the fungal endophyte might be capable of systemic colonization, moving from root to stem. Not all fungal endophytes are capable of systemic colonization; some will remain colonizing at their point of entry (localized colonization). We changed the term vertical transfer to systemic colonization (line 356) as the former denotes different meaning.

Do I understood it correctly that Umbelopsis isabelina … can produce fatty acids that aid in the decaying process.” and therefore represents a “potential element to the decline of Pinus species in Korea?”

Response:

We postulated that the high proportion of saprotrophs and pathotrophs were possibly linked to the decline of pine trees in Korea. We have mentioned Umbelopsis isabelina, together with the other highly abundant saprotroph and pathotroph fungal species to justify their existence and functional classification.

The part with Lophodermium and Rhytismataceae and then of Rhytismatales comprising Rhytismataceae and Lophodermium is confusing.

Response:

Rhytismatales is orders.

Rhytismataceae is family.

Lophodermium is genus.

This is classification. (Eukaryota; Fungi; Dikarya; Ascomycota; Pezizomycotina;Leotiomycetes; Rhytismatales; Rhytismataceae; Lophodermium)

We have clearly indicated the information in the text as follows:

the genus Lophodermium, orders Rhytismatales, The orders Rhytismatales, comprising the family Rhytismataceae

Conclusion

Trivial and oversimplified statements present in this chapter are not based on obtained results, but on textbook facts – fungi have preferences for tissues, are affected by env. factors and needles have the highest abundance of fungal pathogens compared to other parts

Response:

We apologize if you find our conclusions trivial. However, as far as our results are concerned, we have ensured that what we have claimed in our conclusions were appropriately backed up with our findings. We have also ensured that our conclusions tied up with our simple goal of this study. Anyway, we have changed the conclusion section as follows:

Lines 424-432:

We have also shown that some fungal endophytes have preferences in terms of the host and tissues they colonize. Environmental conditions have no significant impact on the overall composition of fungal endophytes in three different tissues of the four Pinus species. We also found that the Pinus species are more prone to needle diseases as demonstrated by the high abundance of plant pathogenic fungi in needle tissue. Moreover, the high proportion of saprotrophs and pathotrophs found in the Pinus species, which implicates a possible emergence of plant disease could be one of the probable causes of the deterioration of Pinus species in Korea.

Reviewer 2 Report

Material and Methods

2.1 Collection of data: I would still need more information and description of the collection. The sample of stem includes: „the bark“ and „central part“, what is central part? Please explain first how needles were sampled, then how stem was sampled and then how roots were sampled. How did you make sure you sampled pine roots?

This sentence: „All tissue samples (needle, stem and root) were processed within 48 h after collection. “ Should be immediately after stating: „All samples were put into clean zip 93 bags separately and kept at 4 °C until further processing.“ As you have to highlight this as storing samples in +4C will not stop the microbiome activity (and in that sense the species composition can change)

2.4 Data analysis. This part needs to be rewritten, it is now difficult to understand. What have been compared and tested?

FunGuild analysis could be improved with manual check oft he results, was this performed?

Line 164: fungal isolates…. You mean species? Or what?

You have 234 different fungal isolates with 88 genera. And then you have 144 operational taxonomic units (OTUs)? Can you explain? What is the definition of OTU? This is very confucing now. Did you sequence 234 isolates and based on IST you grouped them to OTU’s? Which fungi were identified with ITS and which with LSU?

Figure 3 A is not needed, also 3B not needed if these are listed already in Supplementary Table

Line 222: Some endophytes were found only in one tissue, while 31% of all the endophytes (45 OTUs) were found to exist in all three tissues: what are these endophytes? Are they morphologically and molecular ID same? I have difficulties to believe that they actually share same endophytes. Are these penicillium? Are these REAL endophytes???? I would suggest now look into these species and check what literature says. The fact that the real endophytes that are classified in Class 4 by Rodriguez et al in New Phytologist have defined cannot be ignored here. See: Fungal endophytes: diversity and functional roles (2009). If species is found in different tissues it might not be „real“ endophyte. Please check your species

Line 333: Cadophor, should be Cadophora

Still the functions of the fungi are a bit unclear and out of the aim what the results mean? You isolated endophytes. But as said before are these really endophytes or saprotrophs/pathogens having an endophytic life stage. I feel you need to go deeper to each fungus life style

Author Response

Responses to Reviewer 2

Material and Methods

2.1 Collection of data: I would still need more information and description of the collection. The sample of stem includes: „the bark“ and „central part“, what is central part? Please explain first how needles were sampled, then how stem was sampled and then how roots were sampled. How did you make sure you sampled pine roots?

Response:

The sampling protocol has been revised to depict your concerns. We confirmed that the roots were directly connected to pine tree trunk. Part of the protocol goes as follows:

Lines 102-107:

For needle sample, branches were collected from the two opposite sides of the top of the tree. From the collected branches, the needle samples (2-year old) harvested and cut into 0.5 cm segments. Stem samples, one meter above the ground were collected using a sterilized increment borer. The stem samples (15-20 cm) comprised the bark, phloem, sapwood, and heartwood. The root samples were collected from 10 to 25 cm below the ground. The average length of root samples was approximately 30 to 50 cm.

This sentence: „All tissue samples (needle, stem and root) were processed within 48 h after collection. “ Should be immediately after stating: „All samples were put into clean zip bags separately and kept at 4 °C until further processing.“ As you have to highlight this as storing samples in +4C will not stop the microbiome activity (and in that sense the species composition can change)

Response:

I agree with the reviewer. However, there is no other option for preventing the composition change, unless processing the samples in the sampling site. The sentence has been modified as follows:

Lines 109-111:

All samples were put into clean zip bags separately. Samples were transported to the lab in an icebox, kept at 4 °C, and processed within 48 hours.

2.4 Data analysis. This part needs to be rewritten, it is now difficult to understand. What have been compared and tested?

Response:

This section has been modified as follows:

Lines 151-153:

The relative abundance/proportion was estimated as a percentage of the number of fungal species, OTUs or phylum divided by its total number.

FunGuild analysis could be improved with manual check of the results, was this performed?

Response:

We have somehow validated the results from the FunGuild database. We did this through literature search of some previously reported function/lifestyle of the fungi and compared with some representative fungi classified by the FunGuild data.

These statements are as follows:

Lines 415-425:

This can be explained by the high proportion of known saprotrophs Umbelopsis isabellina and Mucor circinelloides in the stem of P. densiflora and P. koraiensis. Umbelopsis isabellina can produce not only protease as wood decaying agent but also can produce fatty acids that aid in the decaying process [48,54]. Mucor circinelloides is used for commercial fermentations and predominantly saprotrophic [55]. All Pinus species, except P. rigida showed the highest fungal endophytes belonging to plant pathogen guild in the needle compared to other tissues. This result suggests that most of the fungi having tendencies to cause diseases were localized in the needle tissues. We further observed that the most abundant fungal strain Septoria sp. in needle has been long known leaf pathogen in plant [56]. Thus, we could speculate that fungal pathogens of pine trees break through the needle tissue and eventually proceed with the infection process.

Line 164: fungal isolates…. You mean species? Or what?

Response:

We apologize for the confusion brought by our statement. We have changed the presentation of the result to convey a more vivid flow of the data. In the new manuscript, we used “isolate” to refer to the total number of culturable fungal endophytes. We changed the result as follows:

Lines 169-183:

We isolated a total of 5,872 culturable fungal endophytes (isolates) from three different tissues of four Pinus species across 18 different sampling sites in Korea. Root tissue presented the largest number of fungal endophytes (n = 2,528) followed by the needle (n = 2,381), and the stem tissues (n = 963). The total number of isolated culturable fungal endophytes varied depending on Pinus species: P. thunbergii (n = 1,791), P. densiflora (n = 1,469), P. rigida (n = 1,379), and P. koraiensis (n = 1,233). From the 5,872 fungal isolates, 234 morpho-species were identified. Further identification of the morpho-species based on rDNA ITS or LSU rDNA revealed 88 genera with a total of 144 operational taxonomic units (OTUs). The sequences were deposited in GenBank (Supplementary Table S2). The fungal endophytes belonged to three known phyla, namely Ascomycota, Basidiomycota, Mucoromycota, representing 91.06%, 5.95% and 2.97%, respectively (Fig.2A). Most of the identified species belonged to Penicillum (14.89%) of Ascomycota, and Umbelopsis (2.12%) of Mucoromycota (Fig.2B). The most frequently occurring phylum Ascomycota, included 127 OTUs. Basidiomycota group was represented by 13 different OTUs. Mucoromycota was represented by 4 OTUs.

You have 234 different fungal isolates with 88 genera. And then you have 144 operational taxonomic units (OTUs)? Can you explain? What is the definition of OTU? This is very confucing now. Did you sequence 234 isolates and based on IST you grouped them to OTU’s?

Response:

Not all the 5,872 isolates were subjected to molecular identification based on ITS/LSU. These 5,872 isolates were initially identified/grouped based on morphological characteristics resulting in the identification of 234 morpho-species. These 234 morpho-species of fungal endophytes were then subjected to molecular identification. After the molecular identification, based on the sequences obtained, we grouped the species into 144 operational taxonomic units (OTUs). In the revised manuscript, we have included a statement in the method section to clearly elaborate this step as well as in the result section.  

Which fungi were identified with ITS and which with LSU?

Response:  

The details were mentioned in Supplementary Table 2.

Figure 3 A is not needed, also 3B not needed if these are listed already in Supplementary Table.

Response:

Only the details on the molecular identification were indicated in the

Supplementary Table2. We included these figures for a clear representation of the proportion of phyla and morpho-species.

Line 222: Some endophytes were found only in one tissue, while 31% of all the endophytes (45 OTUs) were found to exist in all three tissues: what are these endophytes?

Response:

We provided the list in the supplementary data (Table S4).

Are they morphologically and molecular ID same?

Response:

Our definite identification of the fungal isolates was based on molecular typing as it is difficult to designate a proper identification based on morphology alone. However, we ensured that in our initial identification/groupings of the isolates, thorough morphological characterization has been employed.

I have difficulties to believe that they actually share same endophytes. Are these penicillium? Are these REAL endophytes???? I would suggest now look into these species and check what literature says. The fact that the real endophytes that are classified in Class 4 by Rodriguez et al in New Phytologist have defined cannot be ignored here. See: Fungal endophytes: diversity and functional roles (2009). If species is found in different tissues it might not be „real“ endophyte. Please check your species

Response:

As per definition, endophytes are microbes that reside entirely within plant tissues and may grow within roots, stems and/or leaves. We have isolated all the fungi from the tissues of the pine trees, and ensured that surface-dwelling microbes were eliminated in the isolation process as mentioned in the protocol section. We believe that the isolated fungi in this study were indeed, real endophytes.

Line 333: Cadophor, should be Cadophora

Response:

We changed Cadophor to Cadophora.

Still the functions of the fungi are a bit unclear and out of the aim what the results mean? You isolated endophytes. But as said before are these really endophytes or saprotrophs/pathogens having an endophytic life stage. I feel you need to go deeper to each fungus life style

Response:

We isolated many fungal endophytes from pine trees, and most of these fungal endophytes have unknown functions. With an aim of understanding the interaction between the host and fungal endophytes, we have analyzed the possible function of these fungal endophytes. As we have indicated in results section, we found a bunch of saprotrophs and pathotrophs among the isolated fungal endophytes, although this may sound contradictory to the definition of endophytes (as symbionts), still this does not rule out the possibility that these isolates were real endophytes. However, it can be noted that the endophytic lifestyle is common to fungi. Indeed, as you have mentioned, these fungi can be saprotrophs/pathogens having an endophytic stage in their infection process. A statement regarding this speculation has been included in the discussion section. We also indicated their possible implication in the occurrence of disease in pine trees. These statements are as follows:

Lines 367-368:

The endophytic lifestyle of these pathogenic fungi may serve as an entry point for the early stage of the infection process before transitioning to a pathogenic lifestyle.

Lines 431-436:

We also found that the Pinus species are more prone to needle diseases as demonstrated by the high abundance of plant pathogenic fungi in needle tissue. Moreover, the high proportion of saprotrophs and pathotrophs found in the Pinus species, which implicates a possible emergence of plant disease could be one of the probable causes of the deterioration of Pinus species in Korea.
